# Inflammation: Roles in Skeletal Muscle Atrophy

**DOI:** 10.3390/antiox11091686

**Published:** 2022-08-29

**Authors:** Yanan Ji, Ming Li, Mengyuan Chang, Ruiqi Liu, Jiayi Qiu, Kexin Wang, Chunyan Deng, Yuntian Shen, Jianwei Zhu, Wei Wang, Lingchi Xu, Hualin Sun

**Affiliations:** 1Key Laboratory of Neuroregeneration of Jiangsu and Ministry of Education, Co-Innovation Center of Neuroregeneration, NMPA Key Laboratory for Research and Evaluation of Tissue Engineering Technology Products, Jiangsu Clinical Medicine Center of Tissue Engineering and Nerve Injury Repair, Nantong University, Nantong 226001, China; 2Department of Laboratory Medicine, Binhai County People’s Hospital, Yancheng 224500, China; 3Department of Clinical Medicine, Medical College, Nantong University, Nantong 226001, China; 4Department of Orthopedics, Affiliated Hospital of Nantong University, Nantong 226001, China; 5Department of Pathology, Affiliated Hospital of Nantong University, Nantong 226001, China

**Keywords:** skeletal muscle atrophy, inflammation, UPS, ALP

## Abstract

Various diseases can cause skeletal muscle atrophy, usually accompanied by inflammation, mitochondrial dysfunction, apoptosis, decreased protein synthesis, and enhanced proteolysis. The underlying mechanism of inflammation in skeletal muscle atrophy is extremely complex and has not been fully elucidated, thus hindering the development of effective therapeutic drugs and preventive measures for skeletal muscle atrophy. In this review, we elaborate on protein degradation pathways, including the ubiquitin-proteasome system (UPS), the autophagy-lysosome pathway (ALP), the calpain and caspase pathways, the insulin growth factor 1/Akt protein synthesis pathway, myostatin, and muscle satellite cells, in the process of muscle atrophy. Under an inflammatory environment, various pro-inflammatory cytokines directly act on nuclear factor-κB, p38MAPK, and JAK/STAT pathways through the corresponding receptors, and then are involved in muscle atrophy. Inflammation can also indirectly trigger skeletal muscle atrophy by changing the metabolic state of other tissues or cells. This paper explores the changes in the hypothalamic-pituitary-adrenal axis and fat metabolism under inflammatory conditions as well as their effects on skeletal muscle. Moreover, this paper also reviews various signaling pathways related to muscle atrophy under inflammatory conditions, such as cachexia, sepsis, type 2 diabetes mellitus, obesity, chronic obstructive pulmonary disease, chronic kidney disease, and nerve injury. Finally, this paper summarizes anti-amyotrophic drugs and their therapeutic targets for inflammation in recent years. Overall, inflammation is a key factor causing skeletal muscle atrophy, and anti-inflammation might be an effective strategy for the treatment of skeletal muscle atrophy. Various inflammatory factors and their downstream pathways are considered promising targets for the treatment and prevention of skeletal muscle atrophy.

## 1. Introduction

Skeletal muscle is essential for the body’s movement and energy metabolism, accounting for approximately 40% of the total body mass. It is responsible for physical activities and helps people with various daily activities that ensure a good quality of life [1,2]. The mass of skeletal muscle is in a highly dynamic state which depends on a dynamic balance between protein synthesis and degradation. If protein degradation is faster than protein synthesis, skeletal muscle atrophy occurs, resulting in decreased muscle mass and impaired muscle function [3,4], which seriously affects the patient’s daily mobility and living ability.

Skeletal muscle atrophy can be divided into physiological and pathological atrophy. Physiological atrophy occurs mainly in situations of reduced exercise, such as athletes who have discontinued training, those who partake in less physical activity, sedentary people, and astronauts with reduced gravitational load, while pathological atrophy is caused by fasting, nerve damage, stroke, and many other diseases, such as cachexia, diabetes mellitus, and chronic obstructive pulmonary disease (COPD) [3,5,6]. The maintenance of skeletal muscle homeostasis is very important to maintain the normal structure and function of skeletal muscle. Skeletal muscle homeostasis is retained through balancing interactions between some secreted factors and different types of cells in muscle fibers. These cells include muscle satellite cells (MuSCs), motor neurons, and interstitial cells, such as vascular-related cells, fibroblasts, PW1-expressing interstitial cells (PICs), and fibro-adipogenic progenitors (FAPs) [7,8]. Changes in the microenvironment will trigger dynamic changes in the composition of cell types and the interactions between these cells, thereby activating compensatory responses for restoring the original balance in the body [7]. Once the microenvironment continues to change, it will break the skeletal muscle homeostasis, leading to skeletal muscle atrophy.

Our previous studies have shown that skeletal muscle loses its contractile function after denervation, resulting in reduced blood perfusion. Subsequently, a state of hypoxia appears and triggers excessive reactive oxygen species (ROS) production. Consequently, excessive ROS leads to oxidative stress injury as well as a dramatic increase in the production of inflammatory factors, which further activate the inflammatory response pathway and cause inflammation. Over-activated inflammation further initiates the downstream muscle atrophy program with increased proteolysis, decreased synthesis, and reduced muscle regeneration, leading to skeletal muscle atrophy and fibrosis [9,10]. Inflammation and oxidative stress are proposed to be major precipitating factors for skeletal muscle atrophy [11]. Inhibition of inflammation or the inflammatory signaling pathway can markedly suppress denervation-induced inflammatory responses of the skeletal muscle, inhibit the activation of the ubiquitin-proteasome system (UPS) and autophagy-lysosome pathway (ALP), and inhibit mitophagy, thereby alleviating skeletal muscle atrophy [12,13,14]. Our findings further suggest that inflammation plays an important role in the process of denervation-induced muscle atrophy. A variety of inflammation-related diseases are accompanied by skeletal muscle atrophy that seriously affects the prognosis of the primary disease [15,16]. Overall, inflammation is an important contributor to the occurrence and development of skeletal muscle atrophy.

Inflammation is widespread in various diseases, especially in chronic metabolic diseases. Long-term inflammatory stimulation alters the microenvironment of skeletal muscle cells. Notably, these inflammatory cytokines can activate a variety of signaling pathways, including nuclear factor (NF)-κB, Janus-activated kinase/signal transducer and activator of transcription (JAK/STAT), and p38 mitogen-activated protein kinase (MAPK) pathways; destroy the balance between protein synthesis and degradation in the skeletal muscle; enhance skeletal muscle proteolysis; and inhibit protein synthesis, eventually resulting in skeletal muscle atrophy [16,17]. This review clarifies protein degradation and synthesis pathways related to skeletal muscle atrophy, summarizes the mechanism of inflammation in skeletal muscle atrophy, describes the relationship between skeletal muscle atrophy and several inflammation-related diseases (cachexia, sepsis, type 2 diabetes mellitus (T2DM), obesity, COPD, chronic kidney disease (CKD) and nerve injury or nervous system disease), and concludes with some therapeutic strategies for skeletal muscle atrophy by targeting different inflammatory signaling pathways.

## 2. Molecular Mechanisms of Skeletal Muscle Atrophy

Increased proteolysis, decreased protein synthesis, and reduced muscle fiber regeneration are crucial for skeletal muscle atrophy (Figure 1). The activation of proteolytic pathways plays a crucial role in skeletal muscle atrophy. The proteolytic pathways mainly include UPS, ALP, calpains, and caspase-dependent pathways [5,18]. ALP and UPS are mostly responsible for proteolysis and are closely related to skeletal muscle atrophy [2,5,18,19]. Insulin growth factor 1 (IGF-1)/Akt is the main protein synthesis pathway in skeletal muscle [20]. Myostatin (MSTN), a member of the transforming growth factor-β (TGF-β) superfamily, is mainly produced and secreted by skeletal muscle cells and locally restrains muscle growth [21]. MuSCs are major mediators of muscle regeneration when muscle fibers are damaged or stimulated by growth signals [22].

UPS is dramatically activated during skeletal muscle atrophy due to multiple causes, and it is mainly composed of ubiquitin-activating enzyme (E1), ubiquitin-conjugating enzyme (E2), ubiquitin-protein ligase (E3), ubiquitin (Ub), and 26S proteasome [1,23]. The ubiquitination of target proteins refers to the covalent linkage of the target proteins to ubiquitin through a cascade of enzymatic reactions. First, E1 is combined with Ub to form a Ub-E1 complex, and then the Ub-E1 complex binds to E2. During ubiquitination, E2 replaces E1 to form a Ub-E2 complex. Finally, the target protein is transferred to E3 for further processing. All three stages are repeated until the target protein is linked to 4–5 ubiquitin molecules and degraded in the 26S proteasome, resulting in polypeptide hydrolysis, ubiquitin release, and recycling [1,24]. FoxOs are transcription factors that mediate the expression of ubiquitin ligase (E3), and MuRF1 and atrogin-1 are two muscle-specific E3 ubiquitin ligases which play crucial roles in skeletal muscle atrophy. FoxOs are phosphorylated by Akt and consequently enter the cytoplasm from the nucleus to hinder the expression of E3 ubiquitin ligase. When the Akt pathway is blocked or attenuated, FoxOs stay in the nucleus and induce the expression of E3 ubiquitin ligase, activating UPS and causing proteolysis [25,26]. MuRF1 is associated with degradation of myofibrils, myosin light chains 1 and 2, myosin heavy chains, myosin-binding protein C, and troponin 1 [16]. Atrogin-1 is a contributor to vimentin degradation [27]. MuRF1 and atrogin-1 knockout mice are resistant to denervation-induced skeletal muscle atrophy [27]. In addition to MuRF1 and atrogin-1, Casitas B lymphoma-b (Cbl-b) is also a known E3 ubiquitin ligase associated with muscle atrophy. If exercises are reduced or under weightlessness, the expression of Cbl-b increases, ubiquitinating insulin receptor substrate-1 (IRS-1), thereby blocking protein synthesis [28] and further becoming a contributor to skeletal muscle atrophy.

ALP is also an important contributor to skeletal muscle atrophy. Autophagy is a highly conserved homeostatic mechanism that accomplishes the degradation and recycling of cytoplasm, long-lived proteins and organelles through the lysosomal system [29]. Disorders of mitophagy are detrimental to the maintenance of muscle homeostasis, resulting in the accumulation of damaged and dysfunctional mitochondria in cells [30]. In mammals, Parkin, PINK1, and BNIP3 are involved in the regulation of mitophagy. Inactivation of Parkin, PINK1, and BNIP3 can cause mitochondrial abnormalities [4,12,13,19,31]. In addition to mitophagy, other forms of autophagy may play important roles in maintaining skeletal homeostasis. For example, an appropriate nuclear autophagy rate may be extremely important for the nuclear remodeling of myofibers [32]. Three autophagy mechanisms have been discovered in mammals, namely macroautophagy, chaperone-mediated autophagy, and microautophagy. Autophagy in muscle is mostly related to macroautophagy [33]. Macroautophagy plays a very important role in the renewal of myofibrillar proteins, in which autophagosomes are formed to capture protein aggregates or damaged organelles and finally bind to lysosomes for degradation in lysosomes [34]. The mammalian target of rapamycin (mTOR) is an important negative regulator of macroautophagy. Activated mTOR inhibits the key upstream kinase UNC-51-like kinase 1 (ULK1) of macroautophagy through phosphorylation, thereby reducing the activity of macroautophagy. When the ATP level in cells is decreased, AMPK is activated, and the activated AMPK phosphorylates ULK1 to initiate macroautophagy [34,35,36], which thereby plays a vital role in skeletal muscle atrophy.

Calpains mainly include calpain I (CAPN1), calpain II (CAPN2), and muscle-specific calpain (CAPN3) in muscle. CAPN1 and CAPN2 exist as heterodimers consisting of large catalytic subunits and small 30 KDa regulatory subunits. CAPN3 is a classical calpain with a unique structural domain [37]. Calpains are highly regulated and are inactive in most cases. In addition to Ca^2+^-induced conformational changes, calpain activation also requires the hydrolysis of N-terminal or internal peptides. The intramolecular autolysis is accompanied by another structural change, allowing calpains to assemble in the catalytic site and release the active site. CAPN3 is a unique member of the calpain family because it is autolyzed so rapidly that the full-length protein is barely detectable in vitro [38]. Regular exercise training can remodel the skeletal muscle. Due to steric hindrance, macromolecular proteasome complexes cannot directly degrade myofibrils to promote skeletal muscle remodeling, whereas small-molecule calpains can bind to and cleave myofibrils, and myofibrillar proteins can be then released to initiate proteolysis [39,40]. However, these myofibrillar proteins not only can maintain the structural integrity of the myofibrils but are also responsible for force transmission in muscle. Thus, activated calpains can destroy the ultrastructure of myofibers and reduce muscle strength during muscle atrophy [39]. Previous studies have suggested that the major role of calpains is to promote protein turnover by hydrolyzing sarcomeres and cytoskeletal proteins such as titin, dystrophin, nebulin, and desmin [41]. Recently, calpains have been found to cleave specific substrates at specific sites, yielding fragmented proteins with specific biological functions. For example, procaspase-3 is cleaved by calpains to form activated caspase-3, and activated caspase-3 initiates many downstream signaling events [39]. Calpains are also involved in cell signaling by cleaving apoptosis-inducing factors in mitochondria [42]. Thus, calpains not only affect protein turnover but also regulate cell signaling.

Caspases are members of the interleukin-1β converting enzyme family, which are essential in the activation and execution of cell apoptosis and the regulation of inflammation [43]. Caspase-2 is a major apoptotic initiator that activates downstream effector caspases, among which caspase-3 is a key factor in the execution of apoptosis. Caspases are necessary for skeletal muscle differentiation and myogenesis, but the activation of caspases in skeletal muscle differentiation is transient with a lower degree of activation than that observed in apoptotic cells [44]. Caspase-2 is elevated during skeletal muscle cell differentiation, but it is confined to the nucleus throughout the differentiation process and regulates all aspects of the cells that exit from the cell cycle. Therefore, drug inhibition or caspase-2 knockout will markedly alter cell cycle progression and reduce cell differentiation, which is detrimental to skeletal myogenesis [45]. As previously described, as a result of steric, the proteasome cannot directly degrade actin complexes during muscle proteolysis. Caspase-3 is capable of cleaving the actin complex to generate monomeric actin and actin fragments, followed by ubiquitin-proteasome-mediated degradation [46]. In animal models of denervation-induced muscle atrophy, caspase-3 is activated to cause muscle cell death through mitochondria-related apoptosis pathways, thereby promoting muscle atrophy [47,48]. Overall, the caspase family is essential in skeletal muscle differentiation and myogenesis. During muscle atrophy, caspases provide protein substrates for UPS, promote proteolysis, and expedite muscle atrophy through mitochondria-related apoptosis pathways.

IGF-1/Akt is the major pathway for protein synthesis in skeletal muscle. IGF-1 increases protein synthesis in skeletal muscle via the PI3K/Akt/mTOR and PI3K/Akt/GSK3β pathways. PI3K/Akt can also inhibit the transcription of E3 ubiquitin ligase by inhibiting FoxOs, thereby slowing proteolysis [20]. After binding to IGF-1 receptor (IGF-1R), IGF-1 phosphorylates IRS-1, which recruits and phosphorylates PI3K, followed by Akt phosphorylation. mTOR is a downstream substrate of Akt. mTOR has two complexes: mTORC1, including Raptor, which signals to ribosomal protein S6 kinase (S6K) and 4EBP1, controls protein synthesis, and is sensitive to rapamycin; and mTORC2, including Rictor, which signals to Akt but is not sensitive to rapamycin [49]. mTOR1 is considered essential for maintaining muscle mass and functions. Akt activates mTORC1. On the one hand, mTORC1 causes the phosphorylation of p70^S6K^ and the activated p70^S6K^ promotes protein synthesis by activating ribosomal protein S6. On the other hand, mTORC1 also phosphorylates 4EBP1, making its release from the inhibitory complex of eIF4E. Therefore, eIF4E can bind to eIF4G, allowing translation initiation of amino acids [20,50,51]. Glycogen synthase kinase-3β (GSK3β) is another key downstream phosphorylation substrate of IGF-1. The phosphorylated GSK3β is inactivated, resulting in increased activities of eIF2B and β-catenin and promoting protein synthesis [20]. These studies have suggested that IGF-1 increases protein synthesis in skeletal muscle mainly through the PI3K/Akt/mTOR and PI3K/Akt/GSK3β pathways.

MSTN is mainly produced and secreted by skeletal muscle and acts locally in an autocrine or paracrine manner [21]. Evolutionarily, when food is limited, MSTN suppresses muscle hypertrophy, which is an adaptive response in animals to efficiently utilize the limited energy in the body. Therefore, MSTN functions as a negative regulator of skeletal muscle homeostasis. Furthermore, MSTN promotes muscle proteolysis and provides amino acids for hepatic gluconeogenesis in the context of prolonged starvation, enabling the body to survive in harsh conditions [52,53]. On the muscle cell surface, MSTN binds to activin type II B receptor, resulting in the phosphorylation and activation of Smad2 and Smad3 [54]. Phosphorylated Smads enter into the nucleus to regulate the transcription of atrogin-1 and MuRF1, activate UPS, and cause proteolysis [55]. A study found that tumor necrosis factor (TNF)-α can stimulate the expression of MSTN in muscle cells through the NF-κB pathway, and MSTN thereby stimulates the production of interleukin (IL)-6 [56]. Increased IL-6 activates JAK2/STAT3 signaling, which leads to an increased expression of Suppressor of cytokine signaling (SOCS3). SOCS3 inhibits insulin/IGF-1/Akt signaling and p-FoxO by degradation of IRS-1, which results in a reduced protein synthesis and an increased protein degradation, which eventually mediates muscle atrophy. MSTN increases the level of IL-6 via p38MAPK, which corroborates the report that MSTN causes growth inhibition via p38MAPK in muscle cells [57]. In healthy adults, the MSTN-Smad2/3 pathway inhibits the recruitment of MuSCs by activating UPS and caspase-3, thereby inducing proteolysis [58]. Factors such as chronic or acute inflammation, oxidative stress, angiotensin II, and glucocorticoids may all increase MSTN synthesis [54]. Excessive MSTN can over-activate the MSTN-Smad2/3 pathway, which subsequently promotes proteolysis and inhibits protein synthesis, ultimately causing muscle atrophy.

Skeletal muscle has an extraordinary regenerative capacity even if the integrity of fibers is disrupted [59]. Its structure and function will completely recover within several weeks after injury [59]. MuSCs act as the mediators of muscle regeneration, which are partially differentiated quiescent stem cells that live around muscle fibers. If muscle fibers are damaged or receive growth signals, resting-state MuSCs can rapidly re-enter the cell cycle to undergo extensive migration, proliferation, differentiation, and fusion, and form new muscle fibers [22]. The differentiation potential of MuSCs is largely influenced by intrinsic and extrinsic factors. Leukocytes play an important role in regulating the differentiation of MuSCs. Damaged muscle fibers stimulate leukocytes to release inflammatory factors, such as TNF-α and IL-6, which can stimulate MuSCs activation and proliferation [60]. Granulocytes are the first non-resident cells invading damaged muscle, and they have a high phagocytic capacity and play a key role in removing muscle debris. In addition, IL-4 secreted by granulocytes indirectly promotes myogenesis by stimulating FAPs. The pro-inflammatory environment created by granulocytes attracts monocyte recruitment. Classical monocytes secrete higher levels of pro-inflammatory factors, such as TNF-α and IL-1β, to promote myoblast proliferation and delay differentiation. Non-classical monocytes release anti-inflammatory cytokines, such as IL-10 and TGF-β, to reduce myoblast proliferation and stimulate differentiation and fusion [61]. During the middle and late stages of inflammation, partially infiltrating monocytes become mature and differentiate into anti-inflammatory M2 macrophages. M2 macrophages mediate the resolution of inflammation and release IL-4, IL-10, and IGF-1, which also affect myogenesis [62]. IL-4 promotes myotube formation and IGF-1 stimulates myotube hypertrophy [63]. In addition, M2 macrophages secrete large amounts of fibronectin and type VI collagen that will be involved in the self-renewal of MuSCs [22]. In the early stage of muscle injury, the inflammatory response and relevant inflammatory molecules activate and regulate the activation, proliferation and differentiation of MuSCs. During the late stage of muscle injury, the differentiation, fusion, and self-renewal of MuSCs are improved along with the inhibition and recession of the inflammatory response.

## 3. Role of Inflammation in Skeletal Muscle Atrophy

Substantial evidence has shown that inflammation is the driving factor of skeletal muscle atrophy. Inflammatory factors can directly bind to their receptors and activate a series of downstream signaling pathways, suppressing muscle protein synthesis, over-activating proteolysis, and ultimately resulting in skeletal muscle atrophy. In addition to directly acting on muscles, inflammatory factors can also cause metabolic disorders in the digestive system, liver, and other tissues or cells such as adipocytes, thereby indirectly regulating skeletal muscle mass.

### 3.1. Direct Effect: Inflammatory Signaling Pathways Directly Regulate Skeletal Muscle Protein Metabolism

Inflammation directly influences skeletal muscle through the activation of receptor-mediated intramuscular signaling pathways, including the NF-κB, JAK/STAT, and p38MAPK pathways [12,64,65,66].

The NF-κB pathway is a typical pro-inflammatory signaling pathway, and it can promote the expression of inflammatory factors, including cytokines, chemokines, and adhesion molecules. Activation of NF-κB is the key to mediating the entire process of muscle atrophy [67]. NF-κB exists in the form of monodimeric and heterodimeric proteins, and the p50/p65 heterodimer of NF-κB is essential for the transcription of target genes of NF-κB [68]. Generally, the activation of NF-κB is stimulated when pro-inflammatory factors (TNF-α, IL-1) bind to their receptors, ultimately leading to the transcription of target genes, including UPS related molecules, cytokines, chemokines, cell adhesion molecules, and growth factors. These target genes act on UPS, causing proteolysis and triggering skeletal muscle atrophy [9,69]. PGC-1α could enhance the oxidative capacity of skeletal muscle cells, reduce the phosphorylation of NF-κB, suppress the NF-κB pathway, and inhibit the expression of target genes of NF-κB and the generation of pro-inflammatory cytokines, thereby alleviating skeletal muscle atrophy [70]. The importance of NF-κB as a key regulator of muscle atrophy has been highlighted by several in vivo studies. NF-κB-targeted therapy can eliminate muscle atrophy [67], and triptolide can prevent lipopolysaccharide-induced skeletal muscle atrophy by inhibiting the NF-κB/TNF-α pathway [64]. Therefore, the development of anti-muscle atrophy drugs targeting the NF-κB pathway has become the focus of attention.

The JAK/STAT pathway plays a critical role in the regulation of inflammatory responses. It can be activated by type I interferons (IFN-α/β), type II IFNs (IFN-γ), IL-2, and IL-6 [71,72]. IL-6 exerts its biological effects through IL-6 receptor alpha and gp130 [73]. IL-6 binds to p80, which causes gp130 recruitment and homodimerization. The formed protein complex triggers phosphorylation of intracellular tyrosine residues and STAT through Janus kinases (JAK1, JAK2, TYK) [74]. Phosphorylated STATs cause muscle mass loss through different mechanisms. First, muscle proteolysis is activated: STATs enhance the expression of C/EBPδ, which subsequently increases the expression of MSTN, atrogin-1, MuRF1, and caspase-3 in muscle, causing proteolysis through UPS [17,75,76]. Secondly, protein synthesis is inhibited: STAT-C/EBPδ activation increases the expression of MSTN, which inhibits postnatal myogenesis and adversely affects the maintenance of muscle mass [17,76]. Finally, STAT regulates gene transcription by interacting with FOXO and NF-κB [77]. SOCS3, a canonical feedback inhibitor of STAT3 activation, binds to the Tyr759 region of gp130 and inhibits IL-6/gp130 signaling [78]. Overall, the activation of the JAK/STAT pathway plays an important role in skeletal muscle atrophy, and the development of anti-muscle atrophy drugs targeting the JAK/STAT pathway might be greatly valuable in clinical practice.

The major types of MAPKs in mammals include extracellular signal-regulated kinase, c-Jun N-terminal kinase (JNK), and p38MAPK. The JNK and p38MAPK pathways are also known as stress-activated protein kinase pathways. JNK can be activated by upstream MKK4 and MKK7 kinases, and its typical targets are transcription factors such as c-Jun, ATF2, p53, and c-Myc. In addition, JNK can directly phosphorylate other intracellular proteins, such as Bcl-2, Bcl-xL, Bim, and BAD, which are all members of the Bcl2 family and are involved in the regulation of apoptosis [79]. p38MAPK signaling exerts a vital role in the regulation of inflammation and is generally activated by upstream MKK3 and MKK6 kinases [80]. The downstream proteins regulated by p38MAPK can be divided into two categories: transcription factors (such as p53 and ATF2) and protein kinases (such as MAPK2 and MSK1) [80]. p38, a member of the p38MAPK family, can be further divided into p38α (encoded by MAPK14), p38β (encoded by MAPK11), p38γ (encoded by MAPK12), and p38δ (encoded by MAPK13) [80,81]. The p38MAPK pathway is involved in cellular senescence, apoptosis, cell cycle arrest, inflammation, and tumorigenesis. TGF-β-activated kinase 1 (TAK1) usually mediates cytokine receptor-induced p38MAPK activation, and TNF receptor-associated factor (TRAF4) is crucial to the activation of p38MAPK by TAK1. TRAF6 is required for the activation of p38 MAPK by TGF-β receptors [82,83]. p38MAPK is not only a switch for gene expression but also balances the proliferation and differentiation of progenitor cells in response to muscle damage and repairs muscle according to its surrounding environmental stress. p38MAPK has been shown to induce myotube formation during cell fusion. However, p38MAPK activity is excessively enhanced in chronic inflammation, which can cause pathological changes in muscle tissue and muscle atrophy [84]. p38αMAPK is also involved in denervation-induced skeletal muscle atrophy, upregulating MuRF1 and atrogin-1 in muscle cells. CAMK2B is considered a downstream mediator of p38αMAPK, and inhibition of CAMK2B can improve denervation-induced skeletal muscle atrophy [81]. In conclusion, the activation of the p38MAPK pathway exerts an important role in skeletal muscle atrophy, and drug development targeting the p38MAPK pathway might have potential clinical value for the prevention and treatment of muscle atrophy.

### 3.2. Indirect Effects

#### 3.2.1. Inflammation Is Involved in Skeletal Muscle Atrophy via the Hypothalamic-Pituitary-Adrenal Axis

Systemic inflammatory mediators indirectly lead to muscle mass loss through dysregulation of tissues and organ systems. For example, glucocorticoid causes changes in the behavior of the digestive system, liver, and adipocytes through the hypothalamic-pituitary-adrenal (HPA) axis, thereby altering muscle mass [17]. The HPA axis is an important regulatory target that will be activated during systemic inflammatory response, mediating the synthesis and release of endogenous glucocorticoids in the fascicular zone of the adrenal cortex [85,86]. Pro-inflammatory cytokines such as IL-1 and IL-6 can play a role in all parts of the HPA axis to increase the secretion of adrenocorticotropic hormone (ACTH). ACTH is a hormone produced by the pituitary gland and acts as the main physiological regulator of adrenal glucocorticoid secretion. Secreted ACTH binds to MC2R receptors in the adrenal cortex to initiate glucocorticoid synthesis and release [86,87]. Glucocorticoids have been confirmed to protect cells from the deleterious consequences of an overactive inflammatory response. Glucocorticoid receptor (GR) controls the expression of components in two major proteolysis systems (UPS and ALP), as well as the expression of key catabolism-related transcription factors, FoxOs and KLF15 [49,88]. After binding to glucocorticoids, GR translocates into the nucleus and binds to glucocorticoid response elements in the promoter of the target gene, thereby recruiting RNA polymerase II to the nearby transcription start site and activating transcription [49,89]. REDD1 and KLF15 are the target genes of GR in skeletal muscle, which can inhibit mTOR activity. In addition, KLF15 can up-regulate atrogin-1 and MuRF1, and negatively regulate muscle mass [49]. In addition to regulating UPS and FoxOs, glucocorticoids induce resistance to insulin and IGF-1, inhibiting protein synthesis and causing muscle atrophy [90]. Overall, GR leads to muscle atrophy through multiple downstream molecular cascades, but activated mTOR inhibits the transcriptional function of GR and is effectively resistant to glucocorticoid-induced catabolic processes [49]. However, inflammatory cytokines can cause hypothalamic/pituitary negative feedback dysregulation, leading to hyperactivation of the HPA axis, resulting in excessive steroids in the bloodstream, and eventually triggering muscle atrophy [91]. To sum up, the inflammation-HPA axis plays an important role in muscle atrophy, and these studies also provide a new strategy for targeted therapy of muscle atrophy.

#### 3.2.2. Inflammation Is Involved in Skeletal Muscle Atrophy through Fat Metabolism Controls

Studies have shown that inflammatory factors such as IL-6, TNF-α, and IL-1β are related to the control of fat metabolism and muscle mass [92,93]. A single dose of TNF-α could increase muscle proteolysis and lead to anorexia in rats, whereas nutrient intake and body weight were markedly improved in tumor-bearing rats receiving TNF-α inhibitors, which suggests that TNF-α exerts a vital role in the induction of anorexia [17,94]. Reduced food intake or starvation can lead to lipolysis, releasing energy reserves to meet the body’s energy needs. However, catabolism in other tissues will follow as lipid stores are depleted. Hydrolysis of muscle proteins and amino acids is one of the main energy sources in the body [95]. The liver is an important regulator of metabolism, in which amino acids obtained from muscle proteolysis are mainly used for gluconeogenesis and acute phase protein synthesis [96]. In Apc^min/+^ mice with severe cachexia, IL-6 secretion level increased significantly, which inhibited PPAR-α, resulting in hypoketosis and activation of the HPA axis, which ultimately led to increased glucocorticoid release and enhanced muscle proteolysis [97]. The production of PPAR-α-dependent ketones in the liver was impaired, and fenofibrate, a PPAR-α agonist, was used to restore the production of ketone bodies in the liver, thereby reducing the need for gluconeogenesis in the liver and alleviating the need for amino acids from skeletal muscle degradation [98]. These findings indicate hepatic changes in inflammation-related cachexia and the indirect effects of hepatic metabolism and glucocorticoids on skeletal muscle atrophy.

To conclude, inflammation indirectly causes skeletal muscle atrophy via the HPA axis and fat metabolism, and the main mechanism is to regulate the synthesis and release of glucocorticoids and the degradation and synthesis of skeletal muscle proteins. This also provides a new potential target for the prevention and treatment of skeletal muscle atrophy.

## 4. Relationship between Inflammation-Related Diseases and Skeletal Muscle Atrophy

### 4.1. Cachexia and Skeletal Muscle Atrophy

Cachexia is a complex wasting syndrome characterized by systemic inflammatory responses, elevated pro-inflammatory cytokines (IL-1, IL-6, and TNF-α), and remarkable effects on the IGF-1/Akt and NF-κB pathways. IGF-1/Akt is crucial for protein synthesis. When IGF-1/Akt is markedly inhibited by inflammation, the main characterizations include GH resistance and decreased circulating IGF-1 [99]. Local and serum levels of pro-inflammatory cytokines are elevated in cachexia patients, resulting in impaired IGF-1 response to GH, while low levels of IGF-1 are associated with reduced leg muscle cross-sectional area and strength [100]. TNF-α interferes with IGF-1 signaling by inhibiting IRS-1 and IRS-2, downregulating the downstream signaling molecules of IGF-1 involved in the regulation of protein synthesis and cell survival [101]. JNK plays an important role in the inhibition of the IGF-1 pathway by TNF-α. JNK causes phosphorylation of IRS-1 Ser^307^ residues, which separates IRS-1 from the IGF-1 receptor, blocking the downstream signaling of IGF-1 [101]. Whole-transcriptome sequencing analysis of cachectic mice revealed that the NF-κB pathway was dramatically activated [102]. TNF-α and IL-1β activated the NF-κB pathway through the corresponding receptors, and phosphorylated NF-κB entered nucleus, promoting the expression of MuRF1 and atrogin-1, activating UPS, and eventually causing proteolysis. Body mass loss is the prominent hallmark of cachexia in clinical practice, which is induced by two major causes: skeletal muscle atrophy and adipose tissue loss [103]. IL-1β contributes to tumor cachexia mainly through two aspects: one is to trigger an imbalance in skeletal muscle protein synthesis and degradation, and the other is to increase lipolysis, resulting in a chronic stress-like environment, loss of appetite, and increased resting energy expenditure [104]. In conclusion, inflammatory factors are key mediators in the development and progress of cachexia, which will be feasible therapeutic targets for cachexia-induced muscle atrophy.

### 4.2. Sepsis and Skeletal Muscle Atrophy

Sepsis and septic shock are life-threatening diseases, and skeletal muscle atrophy is one of the complications of sepsis. Septic muscle atrophy not only prolongs the time of therapy, but also worsens patients’ prognosis. Patients with sepsis present with increased serum levels of pro-inflammatory cytokines, including TNF-α and IL-6, which increases vascular permeability, induces local tissue edema, slows the flow of capillaries, reduces nutritional supply to muscles, and leads to muscle atrophy [105,106]. Damaged and dysfunctional mitochondria are considerably accumulated, which is an important feature of sepsis-induced muscle dysfunction [107]. Sepsis reduces the activities of complex I and complex IV in the electron transport chain, thereby reducing the ATP/ADP ratio, damaging mitochondria, and promoting the production of ROS [106,108]. Damaged or dysfunctional mitochondria are selectively recycled mainly through mitophagy. Studies have shown that in L6 myotubes treated with septic serum, the expression of autophagy-related proteins—UNC-51, p-Beclin-1, and Beclin-1—and the ratio of LC3B II/I were significantly increased, while the expression of p62 decreased, suggesting that inflammation can over-activate autophagy [109,110]. IGF-1/Akt is a classic signaling pathway that regulates autophagy, and ULK1 is a key protein for initiating autophagy. mTORC1 inhibits the initial step of autophagy by phosphorylating and inactivating ULK1 [35]. Under starvation, AMPK is activated and phosphorylates mTORC1, preventing mTORC1 from phosphorylating ULK1 and leading to the initiation of autophagy [35]. Other cellular stresses, such as ROS accumulation, can activate autophagy by altering the phosphorylation of ULK1 by mTORC1 [111]. mTOR can also regulate the transcription of ULK1 by activating elF4E, 4EBP1, and S6K [112]. Overall, the interaction between the PI3K/AKT/mTOR pathway and autophagy is complex, and mTOR is an important regulator of cell growth and autophagy. Sepsis causes muscle atrophy mainly by regulating autophagy and mitochondrial dysfunction.

### 4.3. Type 2 Diabetes Mellitus and Skeletal Muscle Atrophy

Type 2 diabetes mellitus (T2DM) is characterized by two main features: elevated circulating pro-inflammatory cytokines and free fatty acids, both of which are more pronounced along with the progressive development of hyperglycemia [113]. Insulin resistance in skeletal muscle is greatly harmful to glucose metabolism in vivo, as skeletal muscle is the primary site for glucose absorption and metabolism in response to insulin [114]. Patients with T2DM have higher levels of circulating c-reactive protein (CRP) and various pro-inflammatory cytokines, such as TNF-α and IL-6. Dysregulation of protein metabolism may be the main cause of T2DM-induced skeletal muscle atrophy [115]. A reduction in the sensitivity of mTORC1 to insulin can result in the blockage of the IGF-1/Akt pathway and thereby reduce protein synthesis. Furthermore, in the muscle of obese T2DM patients, a series of pro-inflammatory cytokines are up-regulated, leading to increased proteolysis. Increased TNF-α in T2DM reduces the level of inhibitors of NF-κB (IκB), resulting in hyperactivation of NF-κB and JNK pathways in muscle [116]. Activated NF-κB promotes the expression of ubiquitin ligase and activates UPS to increase proteolysis. JNK mediates insulin resistance and cellular dysfunction, resulting in adipocyte insulin resistance and increased pro-inflammatory cytokines [113]. A high level of IL-6 in muscle can activate the JAK/STAT3 pathway, and phosphorylated STAT can activate UPS, promote proteolysis, and upregulate MSTN, thereby inhibiting myogenesis [117]. In addition, STAT can also interact with the NF-κB pathway to regulate gene transcription. IL-6 also induces SOCS3 expression through the JAK/STAT3 pathway, acting on a negative feedback pathway [118]. Increased SOCS3 impairs glycogen synthesis and glucose transport and inhibits the activity of IRS-1, thereby suppressing the IGF-1/Akt pathway [117]. In addition to skeletal muscle atrophy, T2DM results in the loss of myonuclei in patients. MuSCs dysfunction has been observed in the skeletal muscle of T2DM model animals, which may be attributed to myonuclei loss in T2DM [119]. Pro-inflammatory cytokines, a disturbed IGF-1/Akt pathway, an over-activated inflammatory pathway, and damaged MuSCs are the keys to T2DM-induced skeletal muscle atrophy, but the specific pathogenic mechanism is still unclear and needs further study.

### 4.4. Obesity and Skeletal Muscle Atrophy

Obesity, traditionally defined as an excess of body fat, is a major public health problem. It acts as a key risk factor for the development of T2DM and can induce insulin resistance, which is associated with a series of metabolic disorders, such as dyslipidemia, non-alcoholic fatty liver disease, coronary heart disease, hypertension, and stroke [120,121]. The prevalence of obesity in combination with muscle atrophy is increasing in an ageing population. Studies have shown that obesity promotes the secretion of pro-inflammatory cytokines from adipocytes and macrophages, including TNF-α, IL-1β, and IL-6 [122]. TNF-α is the first cytokine identified to link inflammation and obesity, which is highly expressed in obese people and obese murine models [113]. Under the action of pro-inflammatory cytokines, the unfolded protein response and IKKβ-NF-κB pathways are excessively activated by endoplasmic reticulum stress via the JNK-AP1 pathway, triggering a systemic inflammatory response. Furthermore, systemic oxidative stress, macrophage recruitment, increased NOD-like receptor family protein inflammasome expression, and adipocyte death are major determinants of the pathogenesis of obesity-related inflammation in adipose tissue [123]. Toll-like receptors (TLRs) belong to the IL-1R superfamily that can recognize various microbial pathogen-associated molecular patterns. The TLR4 pathway is a major precipitating factor of obesity-induced inflammation. The activated TLR4 pathway promotes the synthesis of pro-inflammatory cytokines, which initiate hyperactivation of a series of processes downstream of the inflammatory pathway [124]. Obesity-induced muscle atrophy shares partial mechanisms with T2DM-induced muscle atrophy due to the presence of pro-inflammatory cytokines. Therefore, the treatment strategies of the two may also be partially shared.

### 4.5. COPD and Skeletal Muscle Atrophy

COPD is a chronic inflammatory lung disease. Airway inflammation is a consistent feature of COPD, which contributes to persistent airflow limitation and various respiratory symptoms. Diaphragm muscle dysfunction is an important systemic complication of COPD. Multiple studies have shown that skeletal muscle dysfunction in COPD patients is associated with severer airflow obstruction, emphysema, and increased mortality [125,126]. Chronic hypoxia leads to muscle mass loss, possibly as a result of the interaction of multiple molecular mediators, such as inflammation, oxidative stress, decreased oxidase capacity, and a decline in capillary count [127]. IL-6 is highly expressed in COPD patients, which is considered as a biomarker of systemic inflammation. Increased IL-6 is also associated with muscle mass loss and bone loss. In addition, COPD patients exhibit an elevated MSTN in skeletal muscle, while the raised MSTN in the serum of COPD patients is negatively correlated with total muscle mass [128,129]. MSTN can inhibit myoblast proliferation and differentiation, and promote muscle atrophy, which functions as a potent negative regulator of skeletal muscle growth and development [129,130]. After MSTN binds to TGF-β receptor, it activates Smad2 and Smad3 and regulates the transcription of genes involved in the proliferation and differentiation of skeletal muscle precursor cells, as well as the transcription of MuRF1 and atrogin-1 in mature muscle fibers [131]. In addition, MSTN activates the NF-κB pathway and induces an ROS increase in skeletal muscle through the Smad3-NF-κB-TNF-α pathway to cause skeletal muscle atrophy, and inhibits the IGF-1/Akt-mTOR pathway to reduce protein synthesis [129,132]. In COPD patients, MSTN expression is dysregulated, and the overactivated MSTN-Smad2/3 pathway is associated with skeletal muscle dysfunction. In addition, there is no single cellular process leading to muscle atrophy, and multiple disorders are prevalent in COPD patients. Skeletal muscle atrophy induced by COPD is a complex syndrome with multiple etiologies, and the specific molecular mechanisms still need to be further studied.

### 4.6. CKD and Skeletal Muscle Atrophy

Skeletal muscle atrophy is a common and serious complication of CKD. Generally, muscle atrophy caused by CKD is attributed to excessive activation of proteolysis pathways [133]. Potential causes of muscle atrophy in CKD include metabolic acidosis, inflammation, and insulin resistance [134,135,136]. Evidence has suggested that pro-inflammatory factors IL-6, TNF-α, and TLRs can accelerate proteolysis and insulin resistance in CKD [133]. Overexpression of TLR2 and TLR4 can upregulate IL-6, facilitating muscle atrophy [137]. A microarray analysis indicated that TLR13 was upregulated in a mouse model of CKD, possibly leading to the overactivation of the immune system along with insulin resistance. It was also found that TLR13 can interact with interferon regulatory factor 3 (IRF3), resulting in a decrease in Akt phosphorylation, while knockdown of IRF3 prevents TLR13-induced insulin resistance and increases p-Akt [133]. A previous study found that IGF-1 levels seem to be independent of renal function, but IGF-1 binding protein-3 levels increase with the decrease in the glomerular filtration rate, which may lead to a reduced affinity of IGF-1 for IGF-1R in skeletal muscle [138]. The low affinity of IGF-1 for IGF-1R and the reduction in IGF-1 synthesis may impair IGF-1R phosphorylation, leading to skeletal muscle atrophy. Hyperphosphatemia is very common in patients with CKD. A study found that co-culture of L6 cells with high-concentration phosphorus induced cell shrinkage, suggesting that phosphorus overload can directly act on skeletal muscle cells [139]. Hyperphosphatemia in CKD patients may stimulate the autophagy of muscle cells, and the activated autophagy is involved in the subsequent process of muscle proteolysis [139]. Therefore, the over-activation of autophagy cannot be ignored in muscle atrophy in patients with CKD or hyperphosphatemia. Some factors such as insulin resistance and inhibition of the IGF-1/Akt protein synthesis pathway are also necessary for CKD-induced muscle atrophy, which provides a potential scientific target for the development of new drugs for muscle atrophy caused by CKD.

### 4.7. Nerve Injury and Skeletal Muscle Atrophy

Peripheral nerve injury is a common condition in clinics that results in severe atrophy and dysfunction of denervated muscles [140]. Our previous study showed that denervation-induced muscle atrophy can be divided into four distinct transcriptional stages: oxidative stress, inflammation, atrophy, and atrophic fibrosis [9]. This study proposes that inflammation and oxidative stress are major triggers for skeletal muscle atrophy. Interventions for early oxidative stress and inflammatory responses after skeletal muscle denervation can effectively delay the progression of muscle atrophy [4,8,12,13,31,141,142,143]. Madaro et al. also found that denervation-induced muscle damage is a chronic process, and denervation-induced muscle atrophy leads to the progressive accumulation of FAPs, but with no macrophage infiltration and MuSCs-mediated muscle regeneration [7]. FAPs activated by denervation exhibit sustained STAT3 activation and highly secrete IL-6, which further promotes muscle atrophy and fibrosis. Moreover, prolonged denervation leads to a reduction in the number of MuSCs, impairing muscle regeneration after innervation [7,144]. Our previous findings also revealed that inhibition of inflammation (aspirin) or inflammatory signaling pathways (IL-6/JAK/STAT3) could significantly inhibit inflammatory responses in skeletal muscle after denervation, suppress UPS and ALP activation, hinder mitophagy, and then relieve denervation-induced skeletal muscle atrophy [12]. All these findings provide a scientific basis for the development of new drug targets for skeletal muscle atrophy.

Skeletal muscle atrophy caused by central nerve injury is common in spinal muscular atrophy (SMA) and amyotrophic lateral sclerosis (ALS). SMA is a neuromuscular autosomal recessive disorder characterized by progressive degeneration and irreversible damage to lower motor neurons and brainstem nuclei, leading to muscle weakness and atrophy [145]. A mouse model of severe SMA develops systemic inflammation in the early stages and exhibits multiple inflammation-induced responses, including acute-phase responses, overproduction of pro-inflammatory cytokines, and release of glucocorticoids [146]. Systemic inflammation can lead to neurodegenerative changes, including transient and long-term neuronal loss and delayed myelination [146], which provides a reasonable explanation for the sudden onset of SMA that presents as severe multiorgan diseases. ALS is a fatal motor neuron disease with degeneration of motor neurons in the brain motor cortex, brain stem, and spinal cord that disrupts the communication between the nervous system and skeletal muscle, which initially causes muscle weakness and spasm and eventually leads to paralysis in patients along with muscle denervation [147]. Neuroinflammation is increasingly recognized as an important factor for cell-specific neurodegeneration in many central nervous system diseases, including ALS [148]. Upregulation of inflammatory factors has been observed in both ALS patients and animal models, and inflammatory factor levels are closely related to disease severity [149]. Astrocytes and microglia are two major sources of inflammatory mediators in the central nervous system. Activation of NF-κB signaling has been found in several cell types in animal models of ALS. Inhibition of NF-κB in microglia improves motor neuron survival and slows disease progression. Neuron-specific NF-κB inhibition also exhibits beneficial effects [150]. At present, the pathogeneses of SMA and ALS are still unclear. Multiple factors, rather than a single factor, are likely to trigger the occurrence and development of the disease, which makes it difficult to formulate treatment strategies and develop new drugs. Therefore, the research in this field needs to be further strengthened.

## 5. Treatment of Skeletal Muscle Atrophy with Anti-Inflammatory Strategies

Besides drug treatment, there are some other methods including exercise and nutrition support. Exercise is considered the most effective way to treat skeletal muscle atrophy, which promotes blood flow, increases oxygen supply, reduces ROS production, and alleviates inflammatory responses in skeletal muscle [1,151]. However, exercise therapy does not work for everyone. Severely bedridden and neurologically injured patients have apparently limited exercise capacity. The research on the effect of nutrition support on muscle atrophy and clinical outcome of patients with muscle atrophy, especially cachexia and sarcopenia patients, is very limited. Randomized, large-scale, and long-term clinical trials are still needed to test the positive effect of nutritional intervention on muscle metabolism. The combination of multiple methods suitable for individual patients may achieve better therapeutic effects. Anti-inflammatory drugs are a therapeutic direction, and several drugs targeting inflammatory pathways have been applied in basic research on skeletal muscle atrophy. These drugs are mainly used to relieve oxidative stress injury and inflammatory response, which act on inflammatory signaling pathways (including NF-κB, JAK/STAT, and p38MAPK pathways) and delay muscle atrophy (Table 1). For example, conessine inhibits the NF-κB pathway and reduces the nuclear translocation of NF-κB, thereby reducing the expression of MuRF1 and atrogin-1 [152]. Ajoene inhibits the JAK/STAT3 pathway and reduces the production and nuclear translocation of p-STAT3, thereby downregulating the expression of atrophy-related genes [153]. Formononetin reduces the level of inflammatory molecules in serum, relieves muscle damage, protects the PI3K/Akt pathway, and promotes protein synthesis [154]. Astragalus membranaceus and Paeonia japonica inhibit the p38MAPK pathway, thereby suppressing inflammation and alleviating cancer cachexia-induced muscle atrophy [155]. SKP-SC-EVs can mitigate denervation-induced muscle atrophy by inhibiting oxidative stress and inflammation [19]. To conclude, these drugs mainly target inflammatory responses to promote muscle protein synthesis or inhibit proteolysis, thereby alleviating muscle atrophy. These findings further confirm the vital role of inflammation in skeletal muscle atrophy, and also provide a scientific basis for inflammation as a potential target to develop new drugs for the prevention and treatment of skeletal muscle atrophy.

## 6. Prospects

Skeletal muscle atrophy usually occurs with other diseases, leading to inferior quality of life. Inflammation is one that cannot be ignored to induce skeletal muscle atrophy. High levels of pro-inflammatory cytokines result in the disorder of multiple signaling pathways in cells, out-of-control gene expression, and dysfunction of other organs, which triggers the formation of a vicious circle, leads to higher rates of proteolysis, and eventually causes muscle atrophy. The differentiation potential of MuSCs is largely influenced by intrinsic and extrinsic factors. Pathological conditions such as inflammation and oxidative stress in muscle tissue limit the regeneration potential of MuSCs. Consequently, damaged muscle fibers cannot be replenished. A variety of chronic wasting diseases are accompanied by inflammation, and such patients present with varying degrees of skeletal muscle atrophy. However, chronic inflammation-induced skeletal muscle atrophy is a complex process with unclear specific mechanisms that needs further studies. Anti-inflammation is a non-negligible part to delay the procession of muscle atrophy. A variety of anti-inflammatory drugs have been used in basic research. These drugs can promote protein synthesis and inhibit over-activated proteolysis by targeting different inflammatory pathways, ultimately delaying or alleviating skeletal muscle atrophy. Moreover, all anti-inflammatory drugs have shown good therapeutic prospects for delaying muscle atrophy, but further studies on their long-term effects and safety evaluation are still needed before being put into clinical use. In any case, there has been a solid foundation for the development of new drugs that will be used for the prevention and treatment of skeletal muscle atrophy in the future.

## Figures and Tables

**Figure 1 antioxidants-11-01686-f001:**
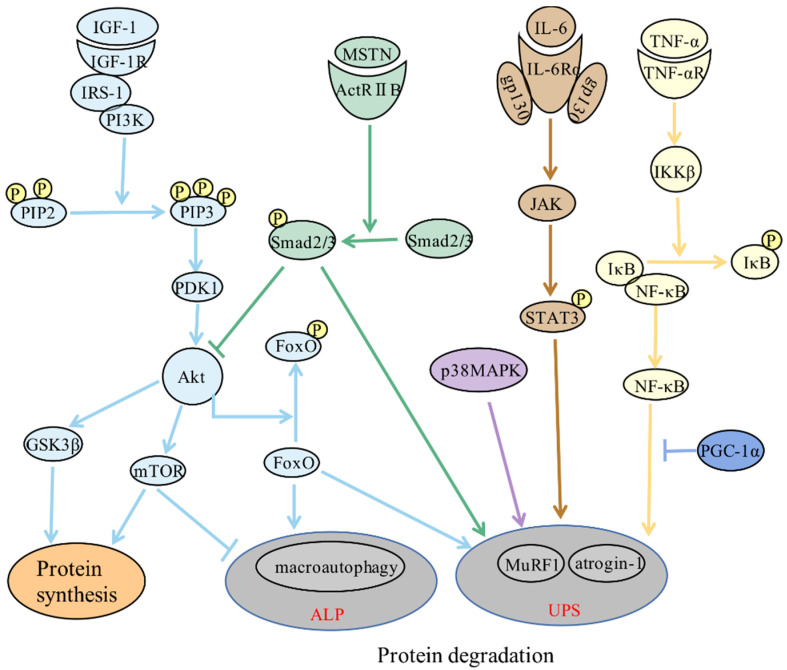
Molecular mechanisms of muscle atrophy.

**Table 1 antioxidants-11-01686-t001:** Drugs targeting inflammation in the treatment of skeletal muscle atrophy.

Drug/Compound	Targeted Signaling Pathway	Function	References
Conessine: a steroidal alkaloid, a potent histamine H3 antagonist	Inhibits NF-κB pathway; inhibits UPS.	Conessine reduces dexamethasone-induced muscle atrophy.	[152]
Salidroside: a glucoside of tyrosol found mostly in the roots of Rhodiola spp.	Inhibits inflammatory response; mitigates oxidative stress; inhibits UPS and mitophagy.	Salidroside alleviates denervation-induced muscle atrophy by suppressing oxidative stress and inflammation.	[4,156]
Carboxyamidotriazole: a noncytotoxic chemotherapy agent, shows anti-inflammatory features	Inhibits inflammation; inhibits NF-κB pathway.	Carboxyamidotriazole can alleviate cancer cachexia-induced muscle wasting.	[157]
Corylifol A: isolated from P. corylifolia	Activates p38MAPK pathway; protects PI3K/Akt pathway.	Corylifol A alleviates muscle atrophy through activating myoblast differentiation and suppressing muscle degradation.	[158]
Lithium chloride	Inhibits inflammatory response; enhances myogenic differentiation; inhibits UPS.	Lithium chloride exerts therapeutic effects on inflammation-mediated skeletal muscle wasting, such as sepsis-induced muscle atrophy and cancer cachexia.	[159]
Fermented oyster extracts (FO): rich in γ-aminobutyric acid (GABA) and lactate	Inhibits inflammatory response; inhibits oxidative stress; inhibits NF-κB pathway; enhances IGF-1/Akt pathway; inhibits UPS.	Fermented oyster extract attenuated dexamethasone-induced muscle atrophy by decreasing oxidative stress and inflammation.	[160]
Liuwei dihuang water extracts: a Chinese herbal medicine composed of Rehmanniae Radix Praeparata, Corni Sarcocarpium, Dioscoreae Rhizoma, Alismatis Rhizoma, Moutan Radicis Cortex and Poria	Attenuates oxidative damage; enhances IGF-1/Akt pathway; inhibits UPS.	Liuwei dihuang water extracts attenuated diabetic muscle atrophy.	[161]
Silibinin (SLI)	Alleviates oxidative stress; inhibits UPS and MSTN; regulates MAPK pathway.	SLI can reduce DDP-induced skeletal muscle atrophy by reducing oxidative stress and regulating ERK/FoxO and JNK/FoxO pathways.	[162]
ATG-125: herbal formula ATG-125, a phytochemical-rich formula	Inhibits chronic inflammation; improves mitochondrial dysfunction; enhances IGF/Akt-mTOR pathway.	ATG-125, as an integrator of protein synthesis and degradative pathways, prevented muscle wasting.	[163]
Leonurus japonicus extract (LJE)	Alleviates inflammatory responses; inhibits NF-κB pathway; inhibits UPS; activates PI3K/Akt pathway.	LJE and leonurine inhibits skeletal muscle atrophy by activating the PI3K/Akt pathway and reducing inflammatory responses.	[164]
Formononetin (FMN): a major bioactive isoflavone compound in Astragalus membranaceus	Inhibits inflammation; inhibits UPS and MSTN; enhances PI3K/Akt/FoxO3a pathway.	Formononetin ameliorates muscle atrophy by regulating myostatin-mediated PI3K/Akt/FoxO3a pathway and satellite cell function in chronic kidney disease.	[154]
BST204: a Rg3 and Rh2 enriched ginseng extract	Inhibits inflammation; reduces oxidative damage; inhibits UPS; activates IGF-1/Akt pathway.	BST204 upregulates myotube formation and mitochondrial function in TNF-α-induced atrophic myotubes.	[165]
Neuregulin-1β (NRG-1β)	Inhibits NF-κB pathway.	Neuregulin-1β modulates myogenesis in septic mouse serum-treated C2C12 myotubes in vitro through PPARγ/NF-κB signaling.	[166]
Resveratrol: a polyphenol that is abundant in grape skin and seeds	Inhibits NF-κB pathway; activates IGF-1/Akt pathway; inhibits UPS and ALP; increases mitochondrial biogenesis.	Resveratrol attenuates skeletal muscle atrophy induced by chronic kidney disease, TNF-α, cancer, and obese muscle atrophy.	[167,168,169]
Carnosol: a bioactive diterpene compound present in Lamiaceae spp.	Inhibits NF-κB and AMPK pathway; activates Akt pathway; inhibits proteolysis.	Carnosol and its analogues attenuate muscle atrophy and fat lipolysis induced by cancer cachexia.	[170]
Histone deacetylase 2	Inhibits NF-κB pathway.	Histone deacetylase 2 suppresses skeletal muscle atrophy and senescence in cigarette-smoke-induced mice with emphysema.	[171]
Ficus carica L.: a flowering plant that contains flavonoids, psoralen, and bergapten	Inhibits the inflammatory response; inhibits NF-κB pathway.	Ficus carica L. attenuates denervated skeletal muscle atrophy via PPARα/NF-κB pathway.	[172]
Targeted ablation of cellular inhibitor of apoptosis 1 (cIAP1)	Inhibits NF-κB pathway; inhibits UPS.	Targeted ablation of cIAP1 attenuates denervation-induced skeletal muscle atrophy.	[173]
P. yezoensis protein (PYCP): from red algae Pyropia yezoensis	Inhibits NF-κB pathway; inhibits UPS and ALP; antioxidant activity.	PYCP protects against TNF-α-induced myotube atrophy in C2C12 myotubes via the NF-κB signaling pathway.	[174,175]
Astragalus membranaceus and Paeonia japonica (APX)	Inhibits inflammatory cytokines; inhibits NF-κB pathway; inhibits p38.	APX protects against muscle atrophy in a C26 colon cancer cachexia mouse model.	[155]
Buyang Huanwu Tang (BYHWT): a compound traditional Chinese herbal medicine	Inhibits inflammatory response; inhibits NF-κB pathway.	BYHWT improves denervation-dependent muscle atrophy by decreasing NF-κB and MuRF1 levels.	[176]
Pyrroloquinoline quinone (PQQ)	Inhibits inflammation; inhibits oxidative stress; inhibits UPS and mitophagy.	PQQ ameliorates skeletal muscle atrophy induced by denervation via inhibition of the inflammatory signaling pathways.	[93,143]
Ajoene: a sulfur compound found in crushed garlic	Inhibits JAK/STAT3 and NF-κB pathway; inhibits SMADs/FoxO pathway; inhibits UPS.	Ajoene extract and Z-ajoene can attenuate skeletal muscle atrophy induced by cancer cachexia through suppressing inflammatory responses.	[153]
Glabridin: a prenylated isoflavone	Inhibits phosphorylation of p38; inhibits FoxO3a.	Glabridin inhibits dexamethasone-induced muscle atrophy.	[177]
Imperatorin (IMP): a main bioactive component of Angelica dahurica Radix	Inhibits JAK/STAT3 pathway; inhibits UPS.	Imperatorin alleviates cancer cachexia and prevents muscle wasting via directly inhibiting STAT3.	[178]
Cryptotanshinone: a major lipophilic compound extracted from the root of Danshen	Inhibits JAK/STAT3 pathway; inhibits UPS.	Cryptotanshinone prevents muscle wasting in CT26-induced cancer cachexia through inhibiting STAT3 signaling pathway.	[179]
Isoquercitrin: a biologically active flavonoid with antioxidative and anti-inflammatory properties	Inhibits inflammatory response; inhibits oxidative stress; inhibits UPS and mitophagy.	Isoquercitrin delays denervated muscle atrophy by inhibiting oxidative stress and inflammation.	[31]
Alantolactone: a sesquiterpene lactone isolated from Inula helenium	Inhibits inflammatory response; inhibits STAT3 pathway.	Alantolactone ameliorates cancer cachexia-associated muscle atrophy mainly by inhibiting the STAT3 signaling pathway.	[180]
S-allyl cysteine: an active component of garlic (Allium sativum)	Inhibits inflammatory response; inhibits NF-κB pathway; inhibits UPS.	S-allyl cysteine inhibits TNFα-induced skeletal muscle wasting through suppressing the expression of inflammatory molecules.	[181]
L-carnitine: antioxidant and anti-inflammatory properties	Reduces the levels of IL-1 and IL-6; inhibits UPS; activates AKT/FOXO3a pathway.	L-carnitine ameliorates the muscle wasting of cancer cachexia.	[182]
Ampelopsin: a natural flavonoid with antioxidant and anti-inflammatory properties	Anti-inflammatory function; anti-oxidative function; inhibits UPS; up-regulates AMPK/SIRT1 pathway.	Ampelopsin attenuates the atrophy of skeletal muscle from d-gal-induced aging rats.	[183]
Aspirin	Inhibits inflammatory response; inhibits JAK2/STAT3 pathway; inhibits UPS.	Aspirin alleviates denervation-induced muscle atrophy via regulating the Sirt1/PGC-1α axis and STAT3 signaling.	[13]
Mito-TEMPO: a mitochondria-targeted antioxidant	Inhibits inflammatory response; ameliorates mitochondrial dysfunction.	Mito-TEMPO improved muscle wasting in CKD mice possibly through inhibiting inflammation and oxidative stress.	[184]
Hemin: an inducer of heme oxygenase-1 (HO-1)	Inhibits proinflammatory cytokine; inhibits ROS; inhibits UPS.	Upregulation of heme oxygenase-1 by hemin alleviates sepsis-induced muscle wasting in mice.	[185]
Avenanthramides (AVAs)	Inhibits inflammatory response; inhibits NF-κB pathway; mitigates oxidative stress; inhibits UPS.	AVAs can reduce proinflammatory cytokine and ROS production and ameliorate TNF-α-induced myotube atrophy in muscle cells.	[186]
SKP-SC-EVs: skin-derived precursors pre-differentiated Schwann cells’ extracellular vesicles	Inhibits inflammation; inhibits oxidative stress; inhibits UPS; inhibits mitophagy.	SKP-SC-EVs mitigate denervated muscle atrophy by inhibiting oxidative stress and inflammation.	[19]

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
