# Peer review of "Inflammation: Roles in Skeletal Muscle Atrophy"

_antioxidants, 2022, doi:10.3390/antiox11091686_

Round 1

Reviewer 1 Report

This excellent review well describes current understanding of the relationship between inflammation and muscle atrophy, which occur associating with various diseases such as cancer (cachexia), diabetes, nerve injury, and so on. Although the mechanisms of muscle atrophy along with chronic diseases are very complex and diverse, the Section 2 explains key players causing atrophy in detail with latest literatures. The Section 3 describes crosstalk between inflammatory factors and atrophic factors. It is one of the main insight of this review; why systemic inflammation accompanies muscle atrophy. The Section 4 introduces basic and clinical findings around muscle atrophy in individual diseases. Since muscle atrophy along with diseases is a serious issue for mortalities of the primary diseases, anti-atrophic therapy would become important in clinical setting in near future. This review provide the current evidences and findings helpful to understand these disease-related muscle atrophy.

Author Response

Thank you for your positive comments about our work.

Reviewer 2 Report

Presently, the ms is overloaded with ideas, concepts and results. My major concern is that the authors are relatively careless to translate experimental data into clinical practice. The ms (including the abstract) would benefit from focussing and a proper stratification of concepts, experimental data and clinical results.

The authors mix up experimental (animal) and clinical data. Ideally, both will add up to allow final conclusions. However, to do so, the authors should demonstrate how the experimental and clinical data add up to each other and how convincing that is. This is not about speculation but about data.

The authors address a no of clinical situations (diseases) assuming that all these diseases result in inflammation (which may look different again in different diseases as well as throughout the course of that diseases) and consequently in a negative energy and nitrogen balance. Faced with the different clinical forms of wasting and malnutrition there is need of a proper differentiation (and stratification of this ms according to individual diseases as well as their staging).

The authors are relatively careless about clinical definitions, e.g., cachexia cannot be explained by a negative energy balance only. That issue is about a negative energy + nitrogen balance. In addition, there are different forms of malnutrition where sarcopenia may be within the narrow focus of this ms. However, to do so a proper definition of sarcopenia as well as the methods used to assess muscle mass in the clinical studies cited have to be described carefully. Or in other words, this is a issue by itself and should not be described circumstantially as part of a review which mainly addresses experimental data.

In some parts of their ms the authors failed to refer to suitable references. E.g., obesity is considered as a low grade inflammatory disease. This general statement cannot be accepted since (i) obesity covers a broad spectrum of body weights and manifestations of co-morbidities, (ii) there are healthy obese subjects and (iii) the issue of sarcopenia obesity refers to a subgroup of obese patients only (e.g., according to the NHANES data the issue is about 15% of the obese subjects). Again, the authors are too fast to draw general conclusions which may fit to their idea rather than to the clinical phenotype.

There are no clinical data (and also no reference to it in this ms) comparing the sarcopenia phenotype associated with obesity with sarcopenia associated with Type 2Dm. This is also true for the individual determinants of sarcopenia (or protein degradation) given in the text of para 4.4.

Although it is likely that anti-inflammatory drugs will have an impact on protein turnover this has to be proven in controlled clinical intervention studies. Until that effects are not proven I consider this idea as hypothetical.

As far as the long list of potential drugs presented in Tab.1 is concerned the authors should add data whether these drugs are already in the pipeline of the industry and if so what is the clinical experience and at which state this process is. In addition, the authors  should add data about possible risks and side effects of the individual drugs.

The authors missed the roles of nutrition and other lifestyle factors in the pathogenesis as well as the treatment of sarcopenia. It is unlikely that sarcopenia can be treated by drugs only. 

Author Response

Presently, the ms is overloaded with ideas, concepts and results. My major concern is that the authors are relatively careless to translate experimental data into clinical practice. The ms (including the abstract) would benefit from focusing and a proper stratification of concepts, experimental data and clinical results.

The authors mix up experimental (animal) and clinical data. Ideally, both will add up to allow final conclusions. However, to do so, the authors should demonstrate how the experimental and clinical data add up to each other and how convincing that is. This is not about speculation but about data.

The authors address a no of clinical situations (diseases) assuming that all these diseases result in inflammation (which may look different again in different diseases as well as throughout the course of that diseases) and consequently in a negative energy and nitrogen balance. Faced with the different clinical forms of wasting and malnutrition there is need of a proper differentiation (and stratification of this ms according to individual diseases as well as their staging).

Response:  We feel great thanks for your professional review work on our article. We are sorry that we mix up experimental and clinical data. We agree that both experimental and clinical data would make the conclusion more convinced. In para 4, when we discuss the relationship between inflammation-related diseases and skeletal muscle atrophy, we elaborate the findings both in basic experimental and clinical study. However, the findings in para 5 are based on experimental data using animals and cells. So far, there are no clinical reports on the direct application of anti-inflammatory related drugs in the treatment of muscle atrophy. The purpose of our article is to elaborate the role of inflammation on muscle atrophy. According to the findings, we speculated that inflammation might be a key factor causing skeletal muscle atrophy. Although all anti-inflammatory drugs have shown good therapeutic prospects for delaying muscle atrophy, further studies on their long-term effects and safety evaluation are still needed before being put into clinical use. Based on your comments, we have made corresponding revisions in the manuscript. And we hope the revised manuscript could be acceptable for you. Moreover, we will refer to your suggestion and make appropriate discussion separately according to different clinical forms of wasting and malnutrition in future articles on drug treatment of muscular atrophy.

The authors are relatively careless about clinical definitions, e.g., cachexia cannot be explained by a negative energy balance only. That issue is about a negative energy + nitrogen balance. In addition, there are different forms of malnutrition where sarcopenia may be within the narrow focus of this ms. However, to do so a proper definition of sarcopenia as well as the methods used to assess muscle mass in the clinical studies cited have to be described carefully. Or in other words, this is a issue by itself and should not be described circumstantially as part of a review which mainly addresses experimental data.

Response:  We feel sorry for our carelessness. We have corrected them and we also feel great thanks for your point out. Cachexia is a complex wasting syndrome, represents the clinical consequence of a chronic, systemic inflammatory response. As there are similar descriptions at the beginning of this paragraph, we made corresponding revisions in the last sentence. We agree with the review that sarcopenia should not be described circumstantially as part of a review. We have replaced “sarcopenia” with “muscle atrophy”.

In some parts of their ms the authors failed to refer to suitable references. E.g., obesity is considered as a low grade inflammatory disease. This general statement cannot be accepted since (i) obesity covers a broad spectrum of body weights and manifestations of co-morbidities, (ii) there are healthy obese subjects and (iii) the issue of sarcopenia obesity refers to a subgroup of obese patients only (e.g., according to the NHANES data the issue is about 15% of the obese subjects). Again, the authors are too fast to draw general conclusions which may fit to their idea rather than to the clinical phenotype. There are no clinical data (and also no reference to it in this ms) comparing the sarcopenia phenotype associated with obesity with sarcopenia associated with Type 2Dm. This is also true for the individual determinants of sarcopenia (or protein degradation) given in the text of para 4.4.

Response: We feel great thanks for your point out. We agree with the reviewer that the issue of sarcopenia obesity refers to a subgroup of obese patients only. Obesity, traditionally defined as an excess of body fat causing prejudice to health, is usually assessed in clinical practice by the body mass index (BMI). The prevalence of obesity in combination with muscle atrophy, a high-risk geriatric syndrome, is increasing in adults aged 65 years and older. Therefore, we have made corresponding revisions in the whole manuscript.

Although it is likely that anti-inflammatory drugs will have an impact on protein turnover this has to be proven in controlled clinical intervention studies. Until that effects are not proven I consider this idea as hypothetical.

As far as the long list of potential drugs presented in Tab.1 is concerned the authors should add data whether these drugs are already in the pipeline of the industry and if so what is the clinical experience and at which state this process is. In addition, the authors should add data about possible risks and side effects of the individual drugs.

Response:Your suggestion really means a lot to us. We agree that the experimental and clinical data adding up to each other would make the conclusion more convinced. In the review, the findings in para 5 are based on experimental data using animals and cells. So far, there are no clinical reports on the direct application of anti-inflammatory related drugs in the treatment of muscle atrophy. However, the purpose of our article is to elaborate the role of inflammation on muscle atrophy. According to the findings,we speculated that inflammation might be a key factor causing skeletal muscle atrophy and anti-inflammation might an effective strategy for the treatment of skeletal muscle atrophy. More studies, including safety evaluation, are needed to select suitable anti-inflammation drugs for clinical use. Therefore, we have made corresponding revisions in the whole manuscript.

The authors missed the roles of nutrition and other lifestyle factors in the pathogenesis as well as the treatment of sarcopenia. It is unlikely that sarcopenia can be treated by drugs only.

Response:Thank you for your constructive suggestion about our work. Besides drug treatment, other methods including physical therapy (electroacupuncture, electrical stimulation, optogenetic technology, heat therapy, low-level laser therapy, etc), nutrition support (protein, essential amino acids, creatine, β-hydroxy-β-methylbutyrate, vitamin D, etc), gene therapy, stem cell and exosome therapy, and other therapies (biomaterial adjuvant therapy, intestinal microbial regulation, oxygen supplementation, etc) might work for the muscle atrophy. However, the purpose of our article is to elaborate the role of inflammation on muscle atrophy. We focused on the therapeutic prospects of drugs related to anti-inflammatory on muscle atrophy. We will summarize the various treatment of muscular atrophy in future’s work. We have made corresponding revision in the manuscript.

Reviewer 3 Report

I am glad the opportunity to review the review article entitled "Inflammation: roles in skeletal muscle atrophy".

This review is well-written and the topics well-organized.

I have minor points that could be included in this review to improve it.

1) It would be great if the authors could add a schematic of proteolytic systems and their interaction with cytokines, and as well as the Akt/Foxo signalling.

2) There are some name initials without the meaning such as CDK, Pi, FAP's. 

3)Please, check the table formatting there are some words that are invading the next column.

4)SOCS-3 is the major regulator of infection and inflammation. However, it is not clear in this review the role of SOCS-3 in muscle atrophy. Does SOCS-3 have an ambiguous role? Does SOCS-3 attenuate or worsen muscle atrophy?

5) Please point out that the diaphragm muscle is the most affected muscle in COPD disease.

Author Response

  • It would be great if the authors could add a schematic of proteolytic systems and their interaction with cytokines, and as well as the Akt/Foxo signalling.

Response:  Thank you for your kind suggestion. We have drawn a schematic and made corresponding revision in the manuscript.

  • There are some name initials without the meaning such as CDK, Pi, FAP's. 

Response:  Thank you for your reminding. We have carefully checked all the initials and made corresponding revisions.

  • Please, check the table formatting there are some words that are invading the next column.

Response:  Thank you for your reminding. We have carefully checked and made corresponding revisions of the table.

  • SOCS-3 is the major regulator of infection and inflammation. However, it is not clear in this review the role of SOCS-3 in muscle atrophy. Does SOCS-3 have an ambiguous role? Does SOCS-3 attenuate or worsen muscle atrophy?

Response: We feel great thanks for your professional review work on our article. We mentioned the role of SOCS-3 in muscle atrophy in many places including Para 2, Para 3.1 and Para 4.3. We are sorry that the role of SOCS3 has not been clarified. Overall, SOCS-3 play a key role in IL6/JAK2/STAT3 pathway mediates muscle atrophy. IL6 acts on myocytes via the gp130 and IL6Rα-complex and activates JAK2/STAT3 signaling, which leads to an increased expression of SOCS3. SOCS3 functions as a negative regulator of cytokine signaling and impairs insulin/IGF-1 signaling by degradation of IRS‐1, that results in a reduced protein synthesis and an increased protein degradation, which eventually mediate muscle atrophy. We have made corresponding revisions.

  • Please point out that the diaphragm muscle is the most affected muscle in COPD disease.

Response: Thank you for your kind suggestion. Diaphragm muscle is the most affected muscle in COPD. We have made corresponding revisions.

Round 2

Reviewer 2 Report

The authors have revised their ms.